# Characterization of Insect–Pollinator Biodiversity in Agrochemical-Contaminated Agricultural Habitats

**Fredrick Ojija** [1,2,*] and **Giovanni Bacaro** [2]

1   Department of Earth Sciences, College of Science and Technical Education, Mbeya University of Science and Technology, Mbeya P.O. Box 131, Tanzania
2   Department of Life Sciences, University of Trieste, Via L. Giorgieri 10, 34127 Trieste, Italy; gbacaro@units.it
*   Correspondence: fredrick.ojija@yahoo.com

**Abstract:** The extensive application of agrochemicals in agricultural habitats in the Southern Highlands of Tanzania (SHOT) is supposed to negatively impact the biodiversity community of insect–pollinators (INPOs). However, in light of existing knowledge, there are no studies to back up this claim. We carried out field surveys in the SHOT to assess and characterize the INPO biodiversity community in agricultural habitats and compare it with protected habitats. Direct observations, transect counts, sweep netting, and pan trap techniques were used for sampling the INPOs. Overall, the INPOs' relative abundance (57.14%) and species diversity index in protected habitats were significantly higher compared to agricultural habitats. Similarly, we recorded a higher number of plant–INPO interactions in protected habitats than agricultural habitats. Our results suggest that, in contrast to protected habitats, agrochemicals might have driven out or discouraged INPOs from agricultural habitats, resulting in dwindling species richness, diversity, and abundance. This could be due to agrochemical contamination that impairs the quantity and quality of floral resources (nectar and pollen) required by INPOs. Alternatively, protected habitats seemed healthy and devoid of agrochemical contamination, which attracted many INPOs for foraging and nesting. Thus, in order to maintain healthy agricultural habitats and support INPO biodiversity, conservation agriculture is imperative.

**Keywords:** agrochemicals; agroecosystem; bees; coleoptera; diptera; food security; lepidoptera; nectar; pollen; pollination

## 1. Introduction

In diverse ecosystems, encompassing agricultural and protected habitats, insect–pollinators (INPOs) are essential invertebrates as they promote biodiversity preservation and the health of ecosystems [1–4]. INPOs are crucial for the pollination services [5,6], as they support the sustainability of countless flowering plants, such as crops, trees, bushes, herbs, vegetables, and many more [7]. They improve agricultural productivity, enhancing human nutrition and food security through pollination services [1]. This implies that a loss of INPOs as pollen transfer vectors may concomitantly lead to a decline in agricultural output, particularly for pollinator-dependent crops [3,8,9]. Despite being vital to life on our planet, compelling evidence from both past and present studies indicates that INPOs are becoming less common in agricultural habitats [3,10]. Apart from a multitude of factors such as exotic invasive species and infections that contribute to their demise or decline [11,12], agrochemicals (Table 1) have also been linked to the decline of INPOs [7,9,13,14]. Increased agrochemical inputs decrease the effectiveness of INPO communities in providing ecological services [10,15,16]. This is because most, if not all, INPOs are very sensitive to exposure to various agrochemicals [7,13,17]. These agrochemicals cover a broad spectrum of chemical agents meant to improve productivity [6,14,18] (Table 1); nevertheless, they are impairing and threatening the INPOs biodiversity community.

**Table 1.** Categories of agrochemicals and their descriptions.

| Agrochemical | Brief Description | Reference |
|---|---|---|
| Pesticides, e.g., insecticides | Synthetic chemicals employed to manage and control pest insects that harm crops. Though they aid in protecting crops from a range of threats that could lower harvests, when wrongly applied, pesticides endanger the INPO biodiversity. Some insecticides affect non-targeted insects, including INPOs, because of their broad spectrum. | [7,10,13,16,19–21] |
| Herbicides | Applied in farms to control and suppress the germination and growth of weeds. They keep fields free of invasive weeds since weeds compete with crops for nutrients, water, light, and space with crops or native plants. But, their incorrect application in agricultural habitats can negatively affect INPOs. | [3,7,13,20,22,23] |
| Fungicides | These are applied to crops to either prevent or treat fungal infections. When imperfectly applied, fungicides pose a danger to INPOs' abundance, richness and diversity. | [7,13,19,20,24,25] |
| Rodenticides | Applied in agricultural habitats to manage rodents (rats and mice) that feast on crops. Like other pesticides, their misuse or inappropriate applications can threaten INPOs. | [23,26] |
| Fertilizers | When synthetic fertilizers are applied on farms, they contaminate plants and flowers, thus reducing forage resource (pollen and nectar) quality and endangering INPOs. | [3,13] |
| Regulators of plant growth | These chemicals are employed to influence growth and development, including plant structure, fruiting, and flowering. Their potential to induce flower abscission can cause the dwindling of INPOs' food resources. | [3,13,14] |

INPOs such as bees, hoverflies, and butterflies could be threatened by the increasing use of harmful agrochemicals (Table 2), such as neonicotinoid pesticides [3,27], which have an adverse impact on non-target species [7,8,10,28]. They can be exposed to agrochemicals in different ways: through air particles, eating contaminated food (nectar or pollen), and consuming contaminated water [4,10,29]. Exposures to agrochemicals could lead to INPOs' mortality [5,9], changes in diurnal activity patterns, e.g., foraging behavior, navigation, and visitation frequency, and other sublethal impacts (Table 2). In addition, it has been shown that a number of agrochemicals, including neonicotinoids, and chlorpyrifos, can severely damage INPOs' nervous systems and irreversibly impair their immune systems, which eventually decreases their biodiversity [15,23,25,30]. Yet, the extent of the negative effects on the INPO biodiversity community of agrochemical exposure has remained unclear in some countries in East Africa, such as Tanzania, due to the paucity of studies and data [26,28,31,32]. Because of the continued, widespread use of perilous agrochemicals, there is a serious threat to the survival of the INPO community in these countries.

**Table 2.** Some examples of the negative effects of agrochemicals on the INPO community in agricultural habitats.

| Impacts on INPOs | Brief Description | Reference |
|---|---|---|
| INPOs' behavior modification | INPOs' navigational, visitation frequency, and foraging abilities can be impaired with by some pesticides,. e.g., bees' lower foraging efficiency and memory impairment have been connected to neonicotinoid insecticides. | [7,10,20,27] |
| Decline in quality and quantity of forage or floral resources | On farms, the quality, diversity and richness of flowering plants can be diminished by the use of synthetic herbicides, insecticides, and other chemicals in agricultural habitats. Some are so offensively scented or contain scents that deter INPOs from visiting the flowers. This reduces the amount of nectar and pollen sources available to INPOs, making it harder for them to obtain enough food. | [7,10,13,20,25,33] |
| INPOs' diseases, loss, and/or death | INPOs are vulnerable to a variety of agrochemicals, i.e., herbicides and insecticides. If contaminated blooming plants are contacted by or consumed by INPOs, the insecticides may kill or cause harm to them. They can also indirectly affect INPOs by lessening the number of flowering plants available for foraging. | [3,7,10,13,20,34] |
| Secondary effects on the food chain | Agrochemical impacts on INPOs might have a cascading or domino impact on the ecosystem. For instance, decreasing and disrupting the INPO biodiversity community could jeopardize other species that depend on INPOs for food and pollination. This is because INPOs are essential for the reproduction of a multitude of plant species, including crops. | [7,10,16,20–22] |
| Drift of pesticide | Agrochemicals, i.e., pesticides or insecticides, can contaminate flowers where INPOs forage when they are carried by the wind or water to undesired ranges. This may cause INPOs' demise due to being exposed to chemicals. | [7,10,13,27,29] |
| Synergistic effects | More adverse effects on INPOs can result from a combination of exposures to multiple agrochemicals (i.e., fungicides, insecticides, and pesticides) than from a single chemical. Multiple agrochemicals combined can impair INPOs' immune systems and make them more vulnerable to infections and diseases. | [7,10,13,24] |
| Sublethal impacts | INPOs can experience harm from pesticide exposure, even when exposed to sublethal levels. Long-term population losses may result from INPOs' reduced ability to navigate, produce offspring, or forage due to exposure to sublethal agrochemical dosages. | [3,7,13,14,20] |

The ongoing extensive application of agrochemicals in agricultural habitats poses a threat to INPOs in the Southern Highlands of Tanzania (SHOT). This is because agrochemicals contaminate and alter the quantity and quality of the floral resources required by INPOs [3,8,35]. The contamination reduces the quality of the nectar, honeydew, and pollen needed by INPOs for energy and nutrients, respectively [13,22]. The loss or decline in the quality of forage resources or niches in contaminated habitats may cause some INPOs to decline or switch to other habitats devoid of agrochemicals [22,34]. This could have a negative impact on the sustainability of the agricultural habitats and nearby ecosystems as a whole [8,34], as well as on the interaction between INPOs and flowering plants, including crops in those habitats [2,20]. The plant–INPO interaction metrics, such as the connectance, linkages per species, generality, linkage density, nestedness, and specialization $H_2'$ index, could be altered in response to change in the INPO biodiversity community and agricultural habitat quality [6,22,36]. In response to changes in the INPO community composition and agricultural habitat quality, the plant–INPO interaction metrics could also be altered or undergo modifications [22,30].

However, it is perplexing to draw conclusions about the overall effects of agrochemicals on INPOs in the SHOT due to the dearth of research on the subject matter, particularly on Tanzania's INPOs biodiversity community. Hence, determining and characterizing the INPOs biodiversity (richness, abundance and diversity) in agricultural habitats and comparing it with protected habitats devoid of agrochemical application could yield information and be used as a baseline to aid in the control of agrochemicals. With this objective, we conducted field surveys in the SHOT, in the Mbeya region, to determine the richness, abundance and diversity of INPOs. Specifically, we hypothesized that (i) agricultural habitats have lower levels of INPO diversity, abundance, and species richness than protected habitats; (ii) agricultural habitats receive less visits from INPOs compared to protected habitats; and (iii) agricultural habitats have lower levels of plant–INPO interactions compared to agricultural habitats.

## 2. Materials and Methods

### 2.1. Study Area

This study was conducted in the Mbeya region's protected habitats (Idugumbi and Loleza forest reserves) and agricultural habitats as well as in their adjacent field margins (Iwambi and Mbalizi), as described in Table 3. From June to October, the Mbeya region experiences cool, dry weather with daily temperatures ranging from 16 to 30 °C [2,12]. From December to May, the region experiences the rainy season, characterized by an average precipitation of 900 mm. Over two million people who reside in the Mbeya region are engaged in agriculture, growing crops and vegetables that depend on INPO pollination, which is their main source of socioeconomic activity [12]. These crops include beans (*Phaseolus vulgaris* L.), watermelon (*Citrullus lanatus* (Thunb.) Matsum. & Nakai) and sunflowers (*Helianthus annuus* L.). But, they also cultivate significant amounts of wind- or self-pollinated crops, such as wheat (*Triticum aestivum* L.), maize (*Zea mays* L.), and rice (*Oryza sativa* L.).

Local farmers use fire to clean the agricultural fields before cultivation. In addition to hand weeding, during the study period, the farmers used post-emergence herbicides to suppress annual grass, sedges, and broad-leaf weeds. For instance, they used Maguguma (Atrazine, S-metolachlor) and Lumax (Mesotrine, Metolachlor, and Trebuthylazine). They also used pesticides such as glyphosate (amino-phosphonates), pyrethroids (lambda-cyhalothrin, cypermethrin), and neonicotinoids (imidacloprids) to control insect pests on the farms. Moreover, the farmers utilized fungicides (e.g., acylalanine and dithiocarbamate) to control fungi in their fields.

**Table 3.** Study sites and their characteristics.

| Habitat Category | Study Site | Habitat Characteristics |
|---|---|---|
| Protected habitats | Loleza (S8.88371, E33.43946) and Idugumbi (S8.88902, E33.33616) forest reserves | Semi-natural habitats are devoid of agrochemicals, e.g., there is no application of herbicides, pesticides, or synthetic fertilizers. Over 50% of these areas are protected forests with permanent grassland and shrublands in the Mbeya Range Forest Reserve. Limited and controlled human activities are allowed, e.g., gathering fuelwood and beekeeping, while cultivation, logging, and grazing are strictly prohibited. The common flowering plant species in these areas include *Lippia kituensis, Leucas grandis, Vernonia galamensis, Aspilia* spp., *Lantana camara, Bidens pilosa, Ocimum gratissimum, Leucas aspera, Leonotis nepetifolia, Ceratotheca* spp., *Tagetes lemmonii, Crotalaria* spp., *Solanum incanum, Lantana viburnoides, Tagetes minuta, Senna didymobotrya, Emilia* spp., *Ageratum conyzoides, Emilia sonchifolia,* and *Commelina benghalensis.* |
| Agricultural habitats | Mbalizi (S8.91788, E33.36285) and Iwambi cultivated farms (S8.92594, E33.37156) | These are agricultural areas where fire is used to clean the field. Also, a wide range of agrochemicals are widely used. Herbicides, insecticides, and fungicides are used to control or suppress weeds, insect pests, and fungi, respectively. Less than 20% of permanent grassland and trees cover these areas; they are unprotected and mostly used for farming and grazing. Examples of plant species in these areas include *Nicandra physaloides, Senna spectabilis, Amaranthus* spp., *Cardopatium* spp., *Abelmoschus esculentus, Cardopatium* spp., *Solanum melongena, Cucurbita pepo, Brassica carinata, Solanum macrocarpon, Phaseolus vulgaris, Solanum lycopersicum,* and *Helianthus annuus.* |

## 2.2. INPOs Sampling

INPOs were sampled in agricultural habitats (Mbalizi and Iwambi) and protected (Loleza and Idugumbi) areas (Table 3). We also sampled INPOs in the field margins near agricultural habitats, as they can also be contaminated by agrochemicals [7,18]. The same size of each habitat was sampled; each was equivalent to $10 \times 10^3$ m$^2$. Pan traps, sweep nets, transect count, and direct observation techniques (Figure 1) were employed in sampling INPOs [1,2,37]. The Mbeya region typically experiences flowering between January and May, which is when the study was carried out. INPO sampling was carried out in 2022 during favorable weather conditions between 8 a.m. and 4 p.m. In every studied habitat, we laid 25 pan traps of blue, white, and UV-reflecting yellow color spaced 5 m apart along 10 transects of 100 m. We placed pan traps during a period of three days per week, from 8 a.m. to 4 p.m. each day. The pan traps were filled with ca. 300 mL of water and 4 mL of scent-free dish soap. INPOs that were captured were collected daily [11,37]. Sweep nets (ca. 35 cm in diameter) were used to record and/or sample INPOs directly [11,37]. Throughout 10 transects of 100 m, the netting was swept at a height of ca. 16 cm above the vegetation and ground layer [37]. Additionally, we used the transect count method over the same transects to count and identify INPOs in the study habitats. Netting and recording were limited to only INPOs that landed on flowers within and along the transects. Sweep netting was carried out twice a week, between 8 a.m. and 11 a.m. and 1 p.m. and 4 p.m. Direct field observations were used to complement each of these methods and complete the list of species. We surveyed the habitats in different days but with standardized weather conditions (full sun and no wind). INPOs were identified at the species or morphospecies level, as well as by their families. Specimens were dried, and others were preserved in 95% ethanol.

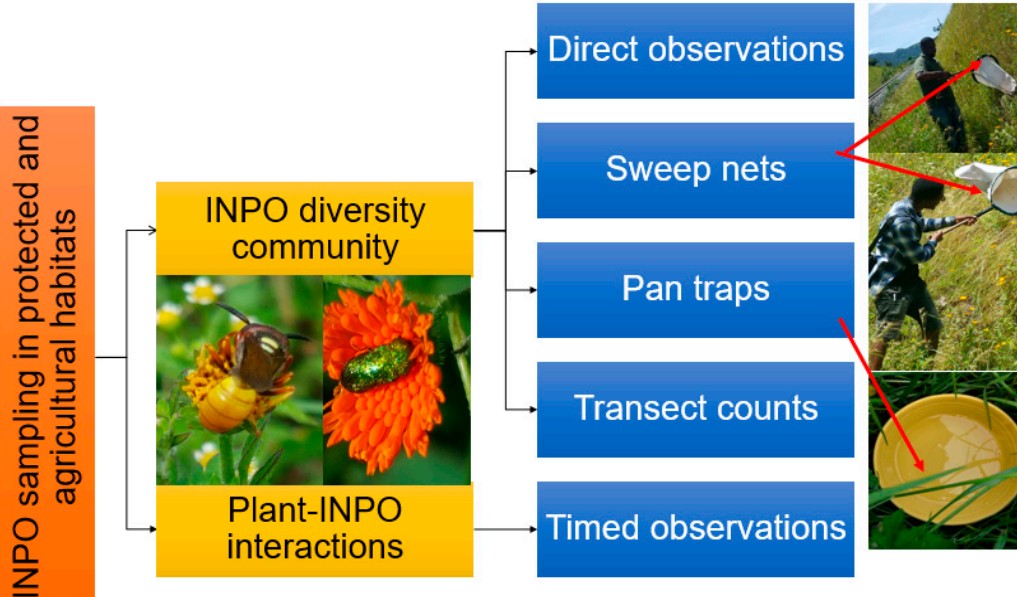

**Figure 1.** An overview of the techniques used to collect INPOs.

### 2.3. Plant–INPO Interaction

To gather data on plant–INPO interactions, timed observations (Figure 1) were carried out at intervals of every 15 min throughout the study habitats twice per week between February and April 2022. The INPOs that were seen by the observers were identified and recorded. Also, we counted the number of INPO visits and identified flowering plant species that received visits. Every week, there were two rounds of observations in each study habitat. INPO observations were consistently conducted on clear, windless days between 8 a.m. and 4 p.m. Both plants and INPOs were identified by species or morphospecies, as well as by their families. The frequency of the INPO visits to flowering plants was used to construct plant–INPO interactions for protected and agricultural habitats. The network-level metrics (connectance, links per species, linkage density, generality, fisher alpha, nestedness, and robustness) were calculated based on the number of INPO visits to each plant species.

### 2.4. Biodiversity Analysis

We pooled data from the four study sites—Loleza and Idugumbi, and Mbalizi and Iwambi—to compare the abundance of INPOs across the two study habitat categories, protected and agricultural habitats, respectively, using the Mann–Whitney U test. The Simpson ($\gamma$), Shannon–Wiener diversity (H′), Evenness (E), Margalef (D), and Fisher alpha ($\alpha$) diversity statistics were also calculated and compared among the study sites. The number of visits by INPOs between protected and agricultural habitats was compared using the Kruskal–Wallis test. It was further used for comparing the abundance across the four study habitats. Species diversity, based on the classic Shannon-Wiener formula, within the four study sites, as well as between the protected and agricultural habitats, was compared using a diversity *t*-test. The normality and homogeneity of variance were confirmed using Levene's and Kolmogorov–Smirnov prior data analyses, respectively. Non-parametric tests (Mann—Whitney U test and Kruskal–Wallis H test) were used when parametric assumptions were not met. Furthermore, the significant differences were verified using Dunn's multiple comparisons test, a post hoc non-parametric test. The significance level for each test was set at five percent (alpha level = 0.05). Paleontological Statistics Software (PAST) version and Origin (9.0 SR1) were used for the various statistical analyses, and R *bipartite* package 2.08 [38] was used to create the plant–INPO webs using R version 3.5.1.

## 3. Results

### 3.1. INPO Sampling and Community Structure

Overall, the INPO abundance in protected habitats was significantly higher (N = 912) than in agricultural (N = 684) habitats (Tables 4 and 5). Agricultural habitats had a lower relative abundance (42.85%) than protected habitats, which had a higher relative abundance of 57.14% (Table 5). Our sampling yield 1596 individuals and 60 species representing 24 families and 5 orders of INPOs (Table 5, Supplementary Materials). Coleopterans (Coccinelidae, Scarabaeidae, and Meloidae), Dipterans (Syrphidae and Muscidae), and Lepidopterans (Pieridae and Nymphalidae) were the most abundant INPOs in protected habitats (Table 5). The most abundant families of INPOs were the Apidae, Nymphalidae, Pieridae, and Syrphidae in protected habitats (Tables 5 and 6). Furthermore, after Hymenoptera, Lepidoptera was the second most common INPO group in protected habitats (Table 6). Compared to the Mbalizi site, the Loleza, Idugumbi, and Iwambi sites exhibited a higher number of INPOs (Table 6).

**Table 4.** Mann–Whitney U test for the statistical testing of the INPO abundance data obtained from the field study carried out in the Mbeya region during the study period.

| **Mann–Whitney U Test** | | |
|---|---|---|
| | **Protected Habitats** | **Agricultural Habitats** |
| Mean rank | 35.39 | 26.12 |
| Standard deviation | 16.72 | 20.34 |
| Standard error | 2.14 | 6.61 |
| U = 1295, $p$ = 0.0038 * | | |

\* Significance difference at $p < 0.05$.

**Table 5.** Order, family, abundance (N), and relative abundance (%) of the INPOs recorded in protected and agricultural habitats in the Mbeya region during the study period.

| Order | Family | **Protected Habitats** | | **Agricultural Habitats** | |
|---|---|---|---|---|---|
| | | **N** | **%** | **N** | **%** |
| Coleoptera | Coccinellidae | 19 | 2.083 | 14 | 2.047 |
| Coleoptera | Meloidea | 31 | 3.399 | 14 | 2.047 |
| Coleoptera | Melyridae | 10 | 1.096 | 2 | 0.292 |
| Coleoptera | Scarabaeidae | 29 | 3.180 | 14 | 2.047 |
| Diptera | Bombyliidae | 21 | 2.303 | 18 | 2.632 |
| Diptera | Calliphoridae | 35 | 3.838 | 25 | 3.655 |
| Diptera | Muscidae | 39 | 4.276 | 33 | 4.825 |
| Diptera | Stratiomyiidae | 14 | 1.535 | 6 | 0.877 |
| Diptera | Tephritidae | 13 | 1.425 | 4 | 0.585 |
| Diptera | Tachinidae | 13 | 1.425 | 5 | 0.731 |
| Diptera | Syrphidae | 45 | 4.934 | 36 | 5.263 |
| Hemiptera | Scutelleridae | 21 | 2.303 | 28 | 4.094 |
| Lepidoptera | Arctiidae | 1 | 0.110 | 0 | 0.000 |
| Lepidoptera | Nymphalidae | 107 | 11.732 | 94 | 13.743 |
| Lepidoptera | Pieridae | 46 | 5.044 | 39 | 5.702 |
| Lepidoptera | Scythrididae | 3 | 0.329 | 1 | 0.146 |
| Hymenoptera | Andrenidae | 6 | 0.658 | 3 | 0.439 |
| Hymenoptera | Apidae | 363 | 39.803 | 298 | 43.567 |
| Hymenoptera | Colletidae | 6 | 0.658 | 0 | 0.000 |
| Hymenoptera | Crabronidae | 3 | 0.329 | 1 | 0.146 |
| Hymenoptera | Formicidae | 28 | 3.070 | 9 | 1.316 |
| Hymenoptera | Halictidae | 16 | 1.754 | 10 | 1.462 |
| Hymenoptera | Megachilidae | 35 | 3.838 | 21 | 3.070 |
| Hymenoptera | Sphecidae | 4 | 0.439 | 0 | 0.000 |
| Hymenoptera | Vespoidae | 4 | 0.439 | 9 | 1.316 |
| Total abundance | | 912 | 100.000 | 684 | 100.000 |

**Table 6.** INPO abundance in each order (**a**) and Dunn's multiple comparison of the overall abundance of INPOs in the study sites (**b**). An asterisk (*) on the *p*-value indicates a significant difference at *p*-value < 0.05.

| (a) | | | | | (b) | |
|---|---|---|---|---|---|---|
| **Order** | **Protected Habitats** | | **Agricultural Habitats** | | **Habitat Comparison** | **p-Value** |
| | **Loleza** | **Idugumbi** | **Mbalizi** | **Iwambi** | | |
| Coleoptera | 58 | 42 | 16 | 29 | Loleza vs. Idugumbi | 0.36 |
| Hemiptera | 7 | 14 | 12 | 16 | Loleza vs. Mbalizi | <0.001 * |
| Diptera | 99 | 68 | 46 | 79 | Idugumbi vs. Mbalizi | <0.001 * |
| Lepidoptera | 75 | 82 | 57 | 77 | Loleza vs. Iwambi | 0.099 |
| Hymenoptera | 244 | 223 | 136 | 216 | Mbalizi vs. Iwambi | 0.006 * |
| Total abundance | 483 | 429 | 267 | 417 | Idugumbi vs. Iwambi | 0.423 |

There was a significant difference (t = 8.42, df = 1254, *p*-value < 0.001) in the species diversity between the protected (H' = 4.379) and agricultural (H' = 3.949) habitats. We further found that protected habitats demonstrated a higher number of species, high Simpson diversity, Shannon–Wiener diversity, Evenness, Margalef, and Fisher alpha diversity index than agricultural habitats (Table 7). In addition, a significant difference in species diversity was observed between and within the study sites (Table 8). However, there was no notable significant difference at the study sites (Loleza and Idugumbi) in the protected areas (Shannon–Wiener diversity, *p*-value = 0.613, and Simpson, *p*-value = 0.938, Table 8). Moreover, significant species diversity was generally observed in protected habitats.

**Table 7.** Biodiversity indices of INPOs in protected and agricultural habitats during the study period.

| **Diversity Indices** | **Protected Habitats** | | **Agricultural Habitats** | |
|---|---|---|---|---|
| | **Loleza** | **Idugumbi** | **Mbalizi** | **Iwambi** |
| Number of species | 54 | 53 | 34 | 49 |
| Simpson ($\gamma$) | 0.96 | 0.96 | 0.91 | 0.94 |
| Fisher alpha ($\alpha$) | 15.58 | 15.91 | 10.34 | 14.42 |
| Evenness (E) | 0.75 | 0.74 | 0.62 | 0.63 |
| Shannon–Wiener diversity (H') | 3.70 | 3.67 | 3.05 | 3.43 |
| Margalef (D) | 8.58 | 8.58 | 5.91 | 7.96 |

**Table 8.** Diversity *t*-test of Shannon and Simpson species diversity for the protected (Loleza and Idugumbi) and agricultural (Mbalizi and Iwambi). An asterisk (*) on the *p*-value indicates a significant difference at *p*-value < 0.05.

| **Habitat Comparison** | **Shannon–Wiener Diversity (H')** | | | **Simpson Diversity** | | |
|---|---|---|---|---|---|---|
| | **t-Value** | **df** | **p-Value** | **t-Value** | **df** | **p-Value** |
| Loleza vs. Idugumbi | 0.506 | 898 | 0.613 | −0.077 | 906 | 0.938 |
| Loleza vs. Mbalizi | 8.215 | 442 | <0.001 * | −3.968 | 318 | <0.001 * |
| Loleza vs. Iwambi | −4.136 | 787 | <0.001 * | 2.872 | 625 | 0.004 * |
| Idugumbi vs. Mbalizi | 7.723 | 460 | <0.001 * | −3.929 | 320 | <0.001 * |
| Idugumbi vs. Iwambi | −3.617 | 792 | <0.001 * | 2.811 | 628 | 0.005 * |
| Mbalizi vs. Iwambi | −4.305 | 558 | <0.001 * | 1.890 | 458 | 0.059 * |

*3.2. Plant–INPO Interactions*

Respectively, the plant–INPO interactions were 1633 and 1385 in the protected and agricultural habitats. The number of interactions was significant (H = 56.64, df = 7, *p*-value < 0.001) between these two habitat categories. While Table 8 shows the interaction metrics of the study habitats, the bipartite graph (Figure 2) illustrates the interactions between flowering plants and INPOs. In both habitat categories, Hymenopterans were the

most active INPOs of flowering plants, followed by Diptera and Lepidoptera (Figure 2). Hymenopterans visited flowering plants in protected habitats more than three times as often as Coleopterans (n = 154, *p*-value < 0.001), with a total of 576 visits. Lepidopterans (*p*-value = 0.030) and Dipterans (*p*-value = 0.560) had more than 237 and 12 visits, respectively (Figure 2). In agricultural habitats, Hymenopterans (n = 511) had 55 more visits than Dipterans (*p*-value = 0.766), twice as many visits as Lepidopterans (n = 281, *p*-value = 0.007), and three times as many visits as Coleopterans (n = 137, *p*-value < 0.001). We observed that not every INPO appeared to visit flowering plants in the same way in each habitat, even though more visitations were recorded in protected habitats (Figure 2). Some crops, *P. vulgaris* and *H. annuus*, in agricultural habitats and surrounding plants, e.g., *B. pilosa* and *Emilia* spp., appeared to alter the INPOs' patterns of interactions in these areas. Moreover, compared to agricultural habitats, protected habitats demonstrated greater levels of connectance, specialization, linkages per species, and nestedness (Table 9).

**Table 9.** Network metrics in the two study areas.

| Network Metrics | Protected Habitats | Agricultural Habitats |
|---|---|---|
| Nestedness | 0.799 | 0.011 |
| Links per species | 3.410 | 3.174 |
| Specialization $H_2'$ index | 0.100 | 0.056 |
| Connectance | 0.974 | 0.961 |
| Generality | 13.86 | 15.656 |
| Linkage density | 8.536 | 9.529 |

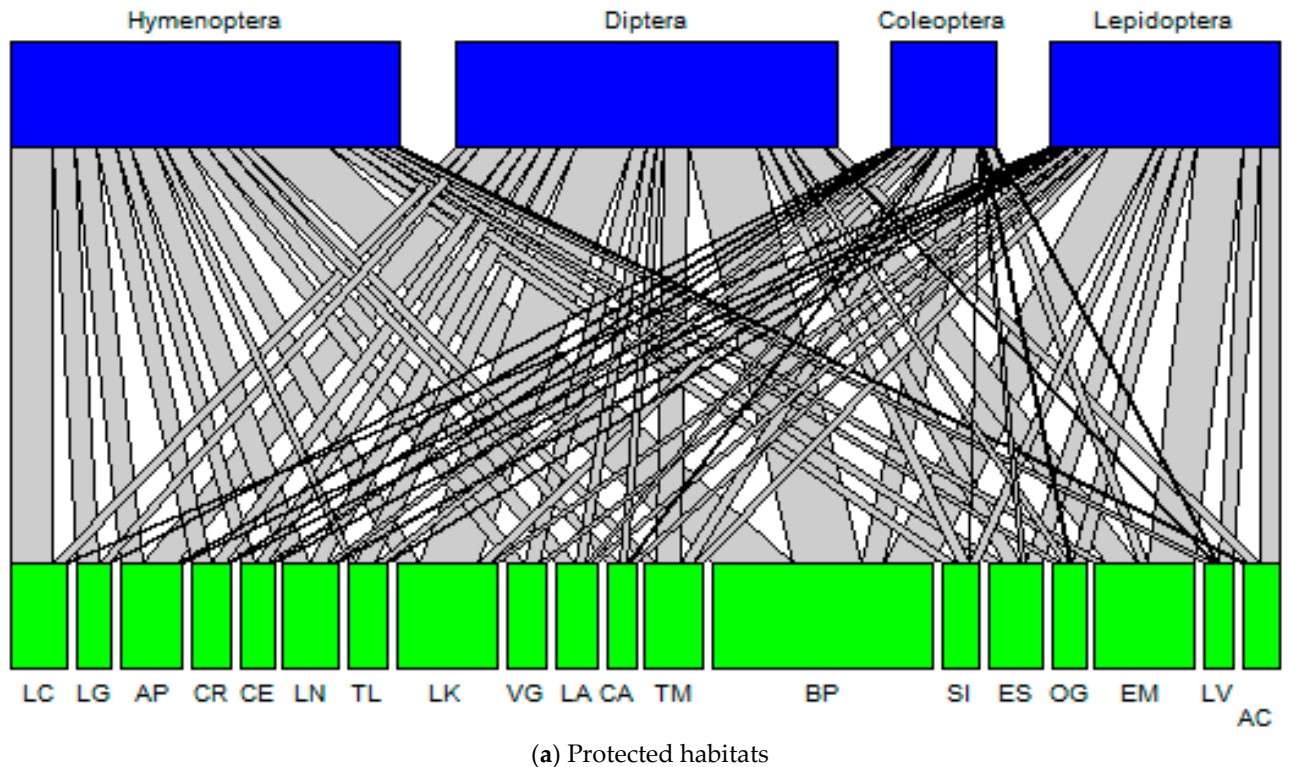

(**a**) Protected habitats

**Figure 2.** *Cont.*

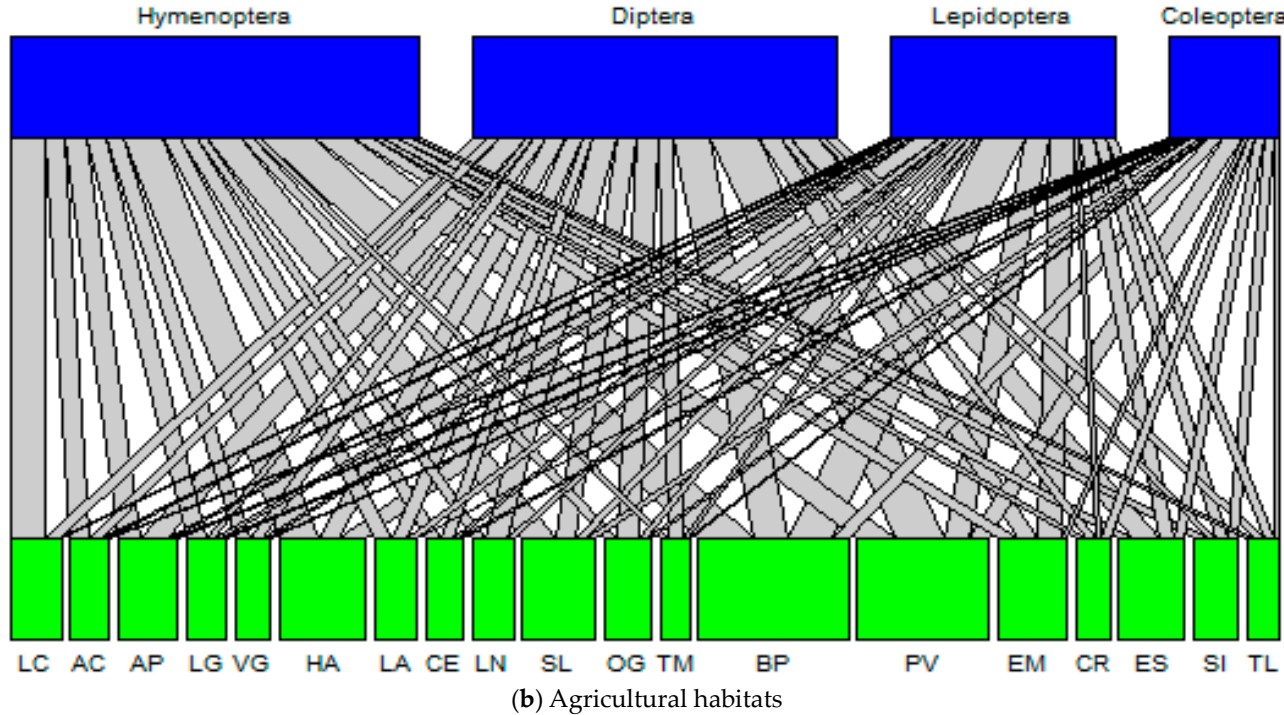

(**b**) Agricultural habitats

**Figure 2.** Plant–INPO interaction web for (**a**) protected and (**b**) agricultural habitats in the Mbeya region in the period of this study. In the web, blue boxes represent INPOs, while green boxes represent flowering plant species. Each box's width represents the total number of visits that were recorded. Plant–INPO interactions are shown by gray links, and the degree of interactions (breadth of the links) indicates the frequency of visits. The abbreviation of plant species are LC = *Lantana camara* L., AC = *Ageratum conyzoides* L. not Hieron, AP = *Aspilia* spp., LG = *Leucas grandis* Vatke, VG = *Vernonia galamensis* (Cass.) Less, HA = *Helianthus annuus* L., LA = *Leucas aspera* (Willd.) Link, CE = *Ceratotheca* spp., LN = *Leonotis nepetifolia* (L.) R. Br, SL = *Solanum lycopersicum*, (L. H. Karst.), OG = *Ocimum gratissimum* L., TM = *Tagetes minuta* L., BP = *Bidens pilosa* L., PV = *Phaseolus vulgaris* L., EM, = *Emilia* spp., CR = *Crotalaria* spp., ES = *Emilia sonchifolia* (L.) DC. ex Wight, SI = *Solanum incanum* L., TL = *Tagetes lemmonii* A.Gray, LK = *Lippia kituensis* Vatke, LV = *Lantana viburnoides* (Forssk.) Vahl, and CA = *Cardopatium* spp.

## 4. Discussion

Our results suggest that the application of agrochemicals has not only a substantial detrimental effect on the INPO biodiversity community but also on their interactions with crops and other flowering plants in agricultural habitats and surrounding field margins. This is because some INPO species are very sensitive to synthetic agrochemicals such as glyphosate and neonicotinoids, neuro-active insecticides [3,7,27]. A study conducted by Main et al. (2020) revealed the presence of neonicotinoids, neuro-active insecticides, in untreated field margins that surround agricultural habitats. Hahn et al. [18] also observed the impact of agrochemicals on INPOs, such as moths, and their role in pollination in the field margin. In general, in agricultural habitats with extensive use of harmful agrochemicals, the abundance of INPOs tends to be quite low [7,18]. Our results provide evidence to back up this claim, as we found low INPO abundance, diversity, and species composition in agricultural habitats where pesticides and other synthetic chemicals (e.g., glyphosate) have been extensively used compared to protected habitats. This indicates that agrochemicals contribute to the decline in INPO biodiversity in many agricultural habitats compared to protected habitats [3,13,39]. The observed decline in INPO biodiversity during our study could be due to the loss of food resources (e.g., pollen and nectar) because of agrochemical contamination [13], as has been reported in other countries [4,14,25,33].

Contamination of flowers reduces the quantity and quality of pollen and nectar [3], thus making it harder for INPOs to survive [20,21]. In support of this, a prior study showed that wildflowers contaminated by fungicides and neonicotinoids reduced the bee abundance in Missouri's conservation areas [7]. Woodcock et al. [4] reported a decline in wild bee species due to contamination by neonicotinoid insecticides. Similarly, Tamburini et al. [25] showed further that exposure to the fungicide azoxystrobin decreased pollen deposition, while exposure to the insecticide sulfoximines reduced bumblebee colony growth, size, and abundance. Moreover, some agrochemicals emit repulsive smells that cause INPOs to withdraw from the agricultural habitats and adjacent fields [3,20]. Owing to this awful scent, most INPOs tend to abandon these habitats and relocate to other suitable areas, i.e., protected habitats, where synthetic chemical application is not practiced [10,21]. Consequently, this leads to a paucity of INPOs (i.e., low abundance, diversity, and species richness) in agricultural habitats [33], as established in our current study. Based on this, the low abundance of INPOs in our agricultural study habitats could be due to pesticide contamination of crops and margin plants in bloom, as well as the awful scents of agrochemicals [13] like glyphosate that were observed applied in our study habitats. Similar observations were also established in previous studies; for example, Main et al. [7] found that the pesticide-contaminated wildflower plants in bloom had a negative effect on the bee abundance. In addition, Sgolastra et al. [14] highlighted that one of the primary causes of the decline in INPOs, such as bees, is agrochemicals.

It appeared that, compared to agricultural habitats, protected habitats devoid of agrochemicals offer abundant and quality floral resources (nectar and pollen), niches, and nesting sites for INPOs [39,40]. As a result, such habitats tend to attract many INPOs, making them more species-rich than polluted habitats with agrochemicals [39,40]. This could also be the reason for the high number of INPO species in our studied protected habitats compared to agricultural habitats during our study. Earlier studies have also demonstrated matching results for the decline in INPO species richness and diversity in agricultural habitats due to agrochemicals [3,14,19]. Furthermore, we found that the INPOs in protected habitats interacted more frequently with flowering plants compared to agricultural habitats. This might be due to the reduced pollen and nectar quality and quantity as a result of agrochemical contamination in agricultural habitats [18,22,36]. The presence of agrochemical residues in pollen and nectar has been reported to cause the low quality of forage resources and a loss of pollinators [14,21]. For instance, Sgolastra et al. [14] and Woodcock et al. [33] claimed the presence of pesticides such as neonicotinoids in pollen and nectar that could decrease INPO abundance and their interactions with plants. Accordingly, we observed lower interaction network metrics in agricultural habitats than in protected habitats. This suggests that the strength of the plant–INPO interaction network depends on the quality of the habitats, forage resources, and surrounding field margins [18,22,36].

It is worth noting that during our study, most farmers were seen spraying various pesticides and herbicides (e.g., glyphosate) within and close to the study habitats. As a result, some INPOs (e.g., hymenopterans, lepidopterans, and dipterans) were seen foraging on flowering plants at the farm margin, and others were ca. 30 to 50 m away from our study habitats at the time farmers were spraying agrochemicals. This would suggest that during the study period, some INPOs altered their foraging activity, including interactions with plants in the study habitats, by relocating away from the contaminated study habitats. A comparable situation was stated regarding the decline in foraging activity of wild bees in the UK on oilseed rape (*Brassica napus* L.) following exposure to neonicotinoids [4]. This is because the visitation pattern of INPOs is influenced by the flower quality, nectar and pollen production [6,18,22,36]. The fewer plant–INPO interactions and lower values of the specialization index in our agricultural habitats might be due to the lower INPO diversity caused by agrochemical contaminations. Since protected habitats had high nestedness, it implies that these habitats had high plant–INPO interactions and were hence more nested. And, high connectance implies resilience and network stability in protected habitats. On the

other hand, fewer plant–INPO interactions in agricultural habitats account for decreased network nestedness and connectance. In general, the difference in the network metrics between the areas found in protected and agricultural habitats is due to the variations in species richness and abundance, as well as floral quality, brought on by the application of agrochemicals [6,18]. Also, preceding studies corroborate our results, as they report a decline in the foraging activity and interaction patterns of INPOs due to exposure to various agrochemicals, i.e., imidacloprid and clothianidin pesticides [6,22,25,36].

## 5. Limitations of the Study

The results and implications of our study only serve as a baseline for future research on the effects of agrochemicals in East and sub-Saharan Africa (e.g., Tanzania, Uganda, Kenya, and Rwanda) and other parts of the world due to a number of limitations, including the lack of an experimental approach and the fact that data were only collected from a limited number of habitats during a single agricultural growing season in the Mbeya region. Additionally, farms that grew mostly beans (*P. vulgaris*) were investigated in this study; thus, different farms with a variety crops applied via agrochemicals need to be studied further. Our results (i.e., abundant INPOs in agricultural habitats compared to protected habitats) may also be limited by the study period. This is due to the fact that during the study period, there was a low abundance of flowering plants influenced by the cultivation techniques, which involved removing or weeding flowering non-crop plant species using hand hoes and/or herbicides, e.g., Maguguma (Atrazine, S-metolachlor). Conversely, if the study was conducted in a period when there was no weeding and clearing of flowering plant species, the results perhaps could be different (i.e., abundant INPOs in protected habitats than agricultural habitats). Thus, it can be assumed that in agricultural habitats, INPO biodiversity could also be influenced by the cultivation techniques (or human disturbance) and differences in local habitat structures, not only by agrochemicals [41,42].

Therefore, it would be vital to conduct additional research—albeit experimental research—to clearly examine how agrochemicals affect INPOs across Tanzania's numerous agricultural habitats, carry out new studies in different periods of cultivation (i.e., before and after weeding or farm cleaning), and compare the INPO biodiversity with protected habitats.

## 6. Conclusions

This study provides a baseline for the state of the research on the impact of extensive agrochemical applications on INPOs in the SHOT. Furthermore, to our understanding, this is the first field study that investigates and presents baseline information on the impacts of agrochemicals in the Mbeya region. The lower number of INPOs in agricultural habitats may indicate that agrochemicals and human-caused changes substantially reduce the quantity and quality of floral resources and the overall health of the ecosystem, which provides the species with feeding and nesting niches. Also, we have shown here the possible effects on pollination services provided by INPOs of both the fatal and sub-lethal consequences of agrochemical application in agricultural habitats. Furthermore, according to our findings, the Mbeya region may experience a decline in INPO biodiversity and pollination services due to the sub-lethal impacts of agrochemicals. Overall, this study is supposed to influence subsequent studies and agrochemical control plans to support the conservation of INPO biodiversity, related food resources and niches, and the health of ecosystems not only in Tanzania but also across East and sub-Saharan Africa.

**Supplementary Materials:** The following supporting information can be downloaded at https://www.mdpi.com/article/10.3390/d16010033/s1, List of species collected.

**Author Contributions:** Conceptualization, F.O.; methodology, F.O.; formal analysis, F.O.; resources, F.O.; data curation, F.O.; writing—original draft preparation, F.O.; writing—review and editing, F.O.; supervision, G.B.; project administration, F.O.; funding acquisition, F.O. and G.B. All authors have read and agreed to the published version of the manuscript.

**Funding:** This research was funded by the British Ecological Society (BES) through the Ecologists in Africa grant (Project EA20/1532) and by the Mbeya University of Science and Technology in Tanzania through an internal research grant (Project No. CoSTE/DAS/2021-2022/01). Moreover, we also thank the Italian Ministry of Foreign Affairs and International Cooperation and the Italian Agency for Development Cooperation for the TWAS-SISSA-Lincei Research Cooperation for funding a research visit that aided in preparing and writing this manuscript.

**Institutional Review Board Statement:** Not applicable.

**Data Availability Statement:** The data presented in this study are available on request from the corresponding author. Data are unavailable due to privacy restrictions.

**Acknowledgments:** We are grateful to Thomas Gervas, Lusekelo Silabi, and Kazumari Mkwavila for their fieldwork support. Furthermore, we would like to thank all of our friends and colleagues who helped us during the design and writing of this article by offering insightful and practical suggestions. We also thank them for taking the time to polish the manuscript and provide an incisive critique that greatly improved the work.

**Conflicts of Interest:** The authors declare no conflicts of interest. The funders had no role in the design of the study; in the collection, analyses, or interpretation of data; in the writing of the manuscript; or in the decision to publish the results.

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
