# Peer review of "Characterization of Insect–Pollinator Biodiversity in Agrochemical-Contaminated Agricultural Habitats"

_diversity, doi:10.3390/d16010033_

Round 1

Reviewer 1 Report

Comments and Suggestions for Authors

The manuscript investigated the difference of biodiversity community of insect pollinators between agricultural habitats and Protected habitats. This will be helpful to comprehend the impact of human agricultural activities on ecosystems. The manuscript could be accepted after some minor revision.

In line 70,it should be table 2.

Author Response

RESPONSE: Thank you, we have made correction. Table 2. Line 69

Reviewer 2 Report

Comments and Suggestions for Authors

The manuscript entitled "Characterization of insect–pollinators biodiversity in agrochem- 2

ical contaminated agricultural habitats", presents an excellent topic, the monitoring of pollinating insects in agroecosystems contaminated with agrachemicals. The introduction is well contextualized, as is the discussion. As a suggestion for improving the quality of the presentation of results, I suggest a better presentation format of the data contained in tables 4 to 9. I suggest that the authors review the flowchart figure, it becomes more visual, to understand the occurrence, species as well as the correlation with the environments.

Author Response

The manuscript entitled "Characterization of insect–pollinators biodiversity in agrochemical contaminated agricultural habitats", presents an excellent topic, the monitoring of pollinating insects in agroecosystems contaminated with agrochemicals. The introduction is well contextualized, as is the discussion. As a suggestion for improving the quality of the presentation of results, I suggest a better presentation format of the data contained in tables 4 to 9. I suggest that the authors review the flowchart figure, it becomes more visual, to understand the occurrence, species as well as the correlation with the environments.

RESPONSE:

Thank you for the suggestion. We have reviewed the tables and no problem with currents formats. We didn’t see any problem with the presentation of the data contained in the tables 4 and 9 as most paper use the same presentation. Also, our presentation is consistent with the objectives. We would kindly to request the current flow of results and tables remained as the same as they will provide readers with good flow of information in the paper.

Reviewer 3 Report

Comments and Suggestions for Authors

The specific manuscript is about the connection between agroecosystems and the presence of insect pollinators, in comparison to wild (non-plant-protection substances) ecosystems in Tanzania. Although the idea is very good, a lot of clarifications are needed.

- At first, I feel that the authors have to clarify if they refer to agroecosystems, or agrochemical contamined ecosystems. There a lot of agroecosystems that they do not received synthetic plant protectants (PP), but they still receive organic PPs and other cultivation techinques, which can affect pollinators. 

- The authors have to clarify about the status of the agroecosystems they examined.  They have to refer all the cultivation techniques the examined fields have delivered (pest/fungi/herb/and all the other -cides application, soil tilleage, weed removal, prunings etc) they have been performed in the crops they sampled.

- In the introduction, I could not find references about the effect of organic farming systems in the presence of pollinators.

- In the M&Ms, a lot of essential info is missing... Please give detailed info of all the cultivation techniques, with emphasis in plant protection applications,, weed control etc, during the study period.

- I could not find any result about the plant biodiversity, which definately affect pollinator presence and biodiversity.

- In the results, the authors talk about proportional abundance. Which are the other insect taxa you added in order to find this proportion? Please give details. Moreover, please provide a detailed table of all the insect species you collected in each habitat.

Specific other comments and suggestions are included in the attached pdf.

Finally, the manuscript will benefit of a linguistic and grammar review.

Comments on the Quality of English Language

The manuscript will benefit of a linguistic and grammar review.

Author Response

The specific manuscript is about the connection between agroecosystems and the presence of insect pollinators, in comparison to wild (non-plant-protection substances) ecosystems in Tanzania. Although the idea is very good, a lot of clarifications are needed. - At first, I feel that the authors have to clarify if they refer to agroecosystems, or agrochemical contained ecosystems. There a lot of agroecosystems that they do not received synthetic plant protectants (PP), but they still receive organic PPs and other cultivation techniques, which can affect pollinators.

RESPONSES:

Thank you for the observation. In our manuscript we referred agroecosystems as agricultural habitats. That is why even the manuscript title is Characterization of insect–pollinators biodiversity in agrochemical contaminated agricultural habitats. Also, the word agricultural habitats has been used mostly in the manuscript. To avoid this ambiguity for readers, the word ‘agroecosystem’ has been replaced by the word ‘agricultural habitats’ throughout the manuscript. Additionally, you are right that some agroecosystems may not receive synthetic plant protectants (PP), but they can still receive organic PPs and other cultivation techniques. However, the current study aimed at assessing the impact of agrochemicals on pollinators, by comparing pollinator community structure between farms (here referred as agroecosystems) and non-farms (here referred to protected habitats).

- The authors have to clarify about the status of the agroecosystems they examined. They have to refer all the cultivation techniques the examined fields have delivered (pest/fungi/herb/and all the other -cides application, soil tilleage, weed removal, prunings etc) they have been performed in the crops they sampled.

RESPONSES:

Thank you for the observation, it is unfortunate that we did not examine this because it was not part of the objectives of the study. We take the comments for improving our future studies. But, we have explained characteristics of the study areas in Table 3.

- In the introduction, I could not find references about the effect of organic farming systems in the presence of pollinators.

RESPONSES:

Yes, we did not include this because we limited ourselves to agrochemicals which is the focus of our study.

- In the M&Ms, a lot of essential info is missing... Please give detailed info of all the cultivation techniques, with emphasis in plant protection applications,, weed control etc., during the study period.

RESPONSES:

We did not collect these information because were not planned in the study objectives. However, we take this observation for our future studies.

- I could not find any result about the plant biodiversity, which definitely affect pollinator presence and biodiversity.

RESPONSES:

All the information and data collected were based on the objectives and planned activities of the study. Unfortunately we did not collect these information because were not planned beforehand in the study objectives. However, we take this observation for improving our future studies.

- In the results, the authors talk about proportional abundance. Which are the other insect taxa you added in order to find this proportion? Please give details. Moreover, please provide a detailed table of all the insect species you collected in each habitat.

RESPONSES:

Actually proportional abundance is the relative abundance. To avoid this ambiguity we have replaced the word ‘proportional abundance’ with ‘relative abundance’. This is shown in Table 5 as percentage (%).

Specific other comments and suggestions are included in the attached pdf.

RESPONSES:

Thank you, we have made some changes and highlighted in yellow

-This is a generalization, since there are a lot of modern selective, narrow spectrum synthetic insecticides, which have very low impact in pollinators and natural enemies. Please rephrase.

RESPONSES:

We have rephrased by replacing the word ‘most’ with ‘some’ Line 48

-In some plant protection products (PPP) you refer only the a.i. (chlorpyrifos) and in others you refer the whole group (neonicotinoid). Since the side effects on beneficial insects is created mostly by the PPP groups, we suggest to use the group name, instead of a single a.i.

RESPONSES:

We have made changes as you suggested. Line 57

-Not only nectar, but honeydew too.

RESPONSES: Honeydew added. Line75

-What was the plant biodiversity within these ecosystems? Please refer. Moreover, please provide a list with the plant species you observe in each ecosystem.

RESPONSES:

We have added the plant species in Table 3

-Glyphosate is herbicide, not pesticide

RESPONSES: We have made correct. Line 367

- Moreover, please provide a detailed table of all the insect species you collected in each habitat.

RESPONSES:

The list of insect pollinator species were attached as Appendix A

Round 2

Reviewer 3 Report

Comments and Suggestions for Authors

After authors' response to our comments, I regret to say that I cannot suggest the manuscript to be published in its current for. Essential info is missing, as we have stated in our past comment, which do not allow to make direct comparison of the pollinators and other insect fauna between the two ecosystems. Although, since the main idea is very good, we suggest the authors to resubmit the manuscript after running another experimental series, without too much variability and statistical 'noise' between the two ecosystems.

Comments on the Quality of English Language

English use is decent, although moderate revision is required.

Author Response

RESPONSE

Thank you very much for the suggestions, and we are sorry that you were not satisfied with our previous responses. The comments you provided have helped to improve our manuscript further and have taught us how to think in a multitude of ways when writing a manuscript. In our study, we opted to use insect pollinators to assess the possible effects of agrochemicals. This is because insect pollinators are widely used as biological indicators of environmental quality. That is why the manuscript mainly focused on insect pollinators alone, as well as comparing them between the two study habitat categories. Our intention was not to compare all the insect fauna in these study habitats. Since other issues were added in the previous revised manuscript as you suggested/recommended, the information that were not fully responded to in the previous revision have been added in the current one, and highlighted in green colour as follows:-

COMMENT:

The authors have to clarify about the status of the agroecosystems they examined. They have to refer all the cultivation techniques the examined fields have delivered (pest/fungi/herb/and all the other -cides application, soil tilleage, weed removal, pruning etc) they have been performed in the crops they sampled.

RESPONSES:

The clarification was added in Table 3. ‘Semi–natural habitats are devoid of agrochemicals e.g., there is no application of herbicides, insecticides, fungicides, and synthetic fertilizers’’, and…… ‘‘These are agricultural areas where fire is used to clean the field. Also, a wide range of agrochemicals are widely used. Herbicides, insecticides, and fungicides are used to control or suppress weeds, insect pests, and fungi, respectively’’.

COMMENT:

In the M&Ms, a lot of essential info is missing... Please give detailed info of all the cultivation techniques, with emphasis in plant protection applications, weed control etc., during the study period.

RESPONSES:

We have added this paragraph..... ‘‘Local farmers use fire to clean the agricultural fields before cultivation. In addition to hand weeding, during the study period, the farmers used post-emergence herbicides to suppress annual grass, sedges, and broad-leaf weeds. For instance, they used Maguguma (Atrazine, S-metolachlor) and Lumax (Mesotrine, Metolachlor, and Trebuthylazine). They also used pesticides such as glyphosate (amino-phosphonates), pyrethroids (lambda-cyhalothrin, cypermethrin), and neonicotinoids (imidacloprids) to control insect pests in the farms. Moreover, the farmers utilized fungicides (e.g., acylalanine and dithiocarbamate) to control fungi in the fields.’’ Line 118-126

COMMENT:

I could not find any result about the plant biodiversity, which definitely affect pollinator presence and biodiversity. RESPONSES: We meant that protected habitats had more abundant and quality floral resources (nectar and pollen) compared to agricultural habitats were pesticides, herbicides, and fungicides are used. We have made this correction in the manuscript. Line 332-334, and Line 356-358 Also, flowering plant species were include in Table 3.

Round 3

Reviewer 3 Report

Comments and Suggestions for Authors

I accept with pleasure the authors responses to my comments, which shows clearly how they believe and support their study. Although, I have to say again that the study has a serious, structural mistake (and some minor others): The authors support that the differences of insect–pollinators biodiversity in their study is coming from the agrochemical contaminated agricultural habitats. This is a true hypothesis, which is based in a totally false methodology.The authors claim that the agroecosystem has less pollinator biodiversity ex officio, comparing with a forest ecosystem. This hypothesis is not supported by a fundamental data, which is the plant biodiversity of the examined ecosystems. Since the pollinators are attracted by plants, the authors have to support their hypothesis that it is the agrochemical use and all the other cultivation technique that they reduce the biodiversity of the plants, the pollinators, or both. Since this kind of data is missing, this claim cannot be supported.  There are other studies which proves that pollinator abundance and biodiversity is higher in agricultural lands compared with jungle environment, due to the higher flowering plant abundance and biodiversity in these lands. 

Concluding, I will advise you again to enrich your study with more data and come back again with your very interesting study. I regret to disappoint  you but this is essential for the robustness of your results.

Best wishes for happy New Year!

Comments on the Quality of English Language

English is satisfying.

Author Response

Thank you for the feedback, we take the comments for improving our future studies. And the missing information we take them as a limitations of our current study.